# Controllable water surface to underwater transition through electrowetting in a hybrid terrestrial-aquatic microrobot

Yufeng Chen [1,2], Neel Doshi [1,2], Benjamin Goldberg [1,2], Hongqiang Wang [1,2] & Robert J. Wood [1,2]

Several animal species demonstrate remarkable locomotive capabilities on land, on water, and under water. A hybrid terrestrial-aquatic robot with similar capabilities requires multi-modal locomotive strategies that reconcile the constraints imposed by the different environments. Here we report the development of a 1.6 g quadrupedal microrobot that can walk on land, swim on water, and transition between the two. This robot utilizes a combination of surface tension and buoyancy to support its weight and generates differential drag using passive flaps to swim forward and turn. Electrowetting is used to break the water surface and transition into water by reducing the contact angle, and subsequently inducing spontaneous wetting. Finally, several design modifications help the robot overcome surface tension and climb a modest incline to transition back onto land. Our results show that microrobots can demonstrate unique locomotive capabilities by leveraging their small size, mesoscale fabrication methods, and surface effects.

[1] John A. Paulson School of Engineering and Applied Sciences, Harvard University, Cambridge, MA 02138, USA. [2] Wyss Institute for Biologically Inspired Engineering, Harvard University, Cambridge, MA 02138, USA. Correspondence and requests for materials should be addressed to Y.C. (email: yufengchen@seas.harvard.edu) or to R.J.W. (email: rjwood@seas.harvard.edu)

Many animal species[1–5] exhibit multimodal locomotive capabilities in terrestrial and aquatic environments to evade predators or search for prey. A few arachnids[3] and insects[4] can move on the surface of water by exploiting static surface tension. The surface tension force significantly exceeds (over 10 times) their body weight, enabling rapid locomotion and even jumping without breaking the water surface[6]. Remarkably some insects, such as diving flies[4] and diving beetles[7,8], can also generate enough force to spontaneously break the water surface to swim, feed, and lay eggs underwater. The ability to move on the water surface, controllably transition into water, and swim underwater enable these creatures to live in complex environments (e.g., high salinity environment such as the Mono Lake[4]) that most animals cannot survive.

Robots that can traverse complex terrains, such as hybrid terrestrial-aquatic environments, are suitable for diverse applications in environmental monitoring and the exploration of confined spaces. Taking inspiration from nature, many robotic prototypes[9–12] have been developed for terrestrial-aquatic locomotion. Most of these amphibious robots, however, weigh over 100 g and cannot move on the water surface due to their large weight-to-size ratio[9–12]. Microrobots (mass<20 g, length<15 cm) have a smaller weight-to-size ratio, and they can leverage surface effects, such as electrostatics or surface tension, to perch on compliant surfaces[13] or move on the surface of water[14–19]. This confers several potential advantages; for example, microrobots can avoid submerged obstacles by returning to the water surface. Furthermore, microrobots experience smaller drag on the water surface due to reduced wetted area, which leads to higher locomotion speed compared to swimming underwater. These hybrid locomotion capabilities could potentially allow microrobots to explore diverse environments that are inaccessible to larger robots.

Previous work[14–20] on water strider-inspired microrobots leveraged insights from studies of biological water striders[21]. These robots weigh 6–20 g, are 8–15 cm long, and use hydrophobic wires for support on the water surface. These devices were used to investigate the biomechanics of water striders[22,23], study the associated fluid mechanics[21], and enable microrobot locomotion on the water surface. Actual water striders are ~1 cm long and weigh 4.5 mg, and they are over 10 times smaller and 1000 times lighter than these microrobots. Since the surface tension force scales linearly with the leg contact length and mass scales with length cubed, these robots need to use supporting legs that are substantially longer than their bodies. These supporting legs create challenges for locomotion in other environments, such as on land or underwater.

In this study, we develop a 1.6 g quadrupedal microrobot that is capable of walking on land, moving on the surface of water, and transitioning between land, the water surface, and aquatic environments (Supplementary Movie 1). We address two challenges that are unexamined in previous studies: gait design for multimodal locomotion[24] in terrestrial and aquatic environments, and strategies for transitions between these environments. First, we develop gaits for locomotion on land and the water surface utilizing a quadrupedal robot with two independently controlled degrees-of-freedom (DOFs) in each leg. Second, we design "feet" that utilize both surface tension and surface tension induced buoyancy to generate the necessary supporting force without inhibiting terrestrial locomotion. The "feet" utilize electrowetting[25–27] to break the water surface. Electrowetting refers to the changing of the liquid to solid surface contact angle in response to an applied voltage, and it is commonly found in microfluidics or electronic paper display applications[26]. We further examine the influence of surface tension on the robot during underwater-to-land transitions. Design changes to the robot's legs and

transmission (compared to a previous version[28]) allow it to overcome a force that is twice its body weight and break the water surface to transition back onto land.

In summary, this work develops multimodal strategies for locomotion in terrestrial and aquatic environments, describes novel mesoscale devices for water surface to underwater transitions, and analyzes the influence of surface tension on microrobot aquatic to terrestrial transitions. These studies culminate in the first terrestrial-aquatic microrobot that adapts to complex environments, representing advances in mesoscale fabrication and microrobot locomotive capabilities.

## Results

**Robot design and demonstration.** We base our robot design on the Harvard Ambulatory MicroRobot (HAMR), which is a 1.43 g quadrupedal robot with eight independently actuated DOFs[29]. Two piezoelectric actuators control the swing (fore/aft) and lift (vertical) motion of each leg. The robot is fabricated based on the PC-MEMS process[28,30] and the robot transmissions are made of compliant 25 μm polyimide flexures (Kapton, Dupont). In previous studies[28,29], a number of walking gaits such as the trot, pronk, and jump are shown to be capable of high speed locomotion on land.

Several design modifications are implemented (Fig. 1a) to enable locomotion on the water surface, controllable sinking, and transitions from underwater to land. The legs are equipped with passive, unidirectional flaps to facilitate swimming (Fig. 1b), and with electrowetting pads (EWP-Fig. 1c) to generate surface tension and buoyant forces to support the robot's weight on the water surface. To break the water surface and transition into water, the EWPs utilize electrowetting to modify surface wettability. In addition, design modifications are made to the robot's chassis and circuit boards to reduce the volume of air trapped during sinking. To avoid shorting underwater, the circuitry is coated in ~10 μm of Parylene C. Finally, the robot transmissions are manually stiffened approximately two times to improve vehicle payload, which allows a submerged robot to break the water surface and transition back to land.

This robot can walk on level ground, transition from ground onto the water surface, swim on the water surface to evade underwater obstacles, sink into water by actuating its EWPs, walk underwater, and transition back onto land by climbing an incline (Fig. 2a, and Supplementary Movie 1). Figure 2b shows a corresponding experimental demonstration of robot locomotion. The robot moves at a speed of 7 cm s$^{-1}$ using a 10 Hz trot gait on level ground. To walk from land onto the water surface (Fig. 2c, and Supplementary Movie 2), the robot walks down a 7° incline using a trot gait with a 1 Hz stride frequency to avoid breaking the water surface in this process. Once the robot is afloat, it swims (Fig. 2d, and Supplementary Movie 3) at a speed of 2.8 cm s$^{-1}$ using a 5 Hz swimming gait. To dive into water, the actuators are switched off and a 600 V signal is applied to the EWPs. The locally induced electric field modifies the surface wettability, reduces the surface tension force, and causes the robot to sink into water (Supplementary Movie 4). Once the robot sinks to the bottom, it can walk underwater (Fig. 2e, Supplementary Movie 5) using a trot gait at stride frequencies up to 4 Hz. To transition back onto land, the robot climbs an incline of up to 6° and gradually moves through the water-air interface (Fig. 2f, g, and Supplementary Movie 6). Further, the robot can demonstrate turning on land, on the water surface, and underwater to avoid obstacles. In the following sections, we describe detailed results on robot swimming and transition between land, the water surface, and underwater environments.

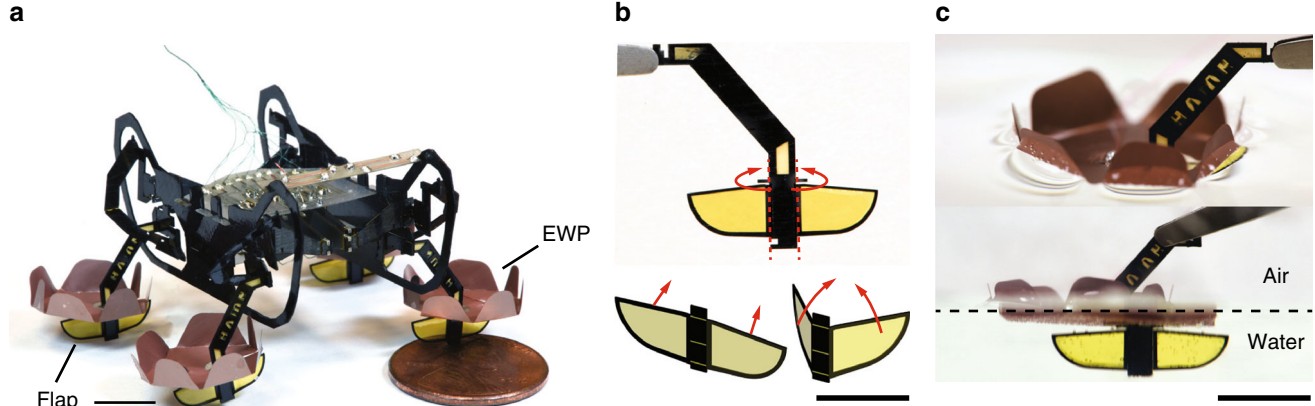

**Fig. 1** Design of a hybrid terrestrial-aquatic microrobot and its electrowetting pads. **a** A quadrupedal, 1.6 g, 4 cm × 2 cm × 2 cm hybrid terrestrial-aquatic microrbot. The robot is powered by eight piezoelectric actuators and each leg has two independent degrees-of-freedom. Each robot leg consists of an electrowetting pad (EWP) and two passive flaps. **b** Two passive flaps are connected to the central rigid support via compliant polyimide flexures. These passive flaps retract under drag forces opposing the robot's heading but remain open under thrust forces in the same direction as the robot's heading. **c** Perspective and front views of an EWP on the water surface. The EWP supports the robot weight via surface tension effects and the flaps paddle underwater to generate thrust forces. Scale bars (**b–c**), 5 mm

**Floating and controllable sinking through electrowetting.** One of the major challenges in developing a robot capable of moving on the water surface is to support the robot's weight. Previously developed water strider-inspired robots[5,16,18] are 10 times larger and 100–1000 times heavier than natural water striders, and they rely on multiple, non-moving legs to support themselves on the water surface. Such a design limits these robots' ability to move through cluttered environments and to traverse other types of terrains. Here, we develop a novel design that substantially reduces leg length, enables controllable sinking through electro-wetting, and allows for multimodal locomotion on land, on the water surface, and underwater.

An EWP (Fig. 3a) is installed on each leg of HAMR. The EWP is ~1 cm in diameter, and it is made of a folded 5 μm thick copper sheet coated by 15 μm Parylene (see Methods). This hydrophobic, dielectric coating insulates the copper from water. Unlike previous water surface supporting devices that only utilize surface tension, our device relies on surface tension and surface tension induced buoyancy. The maximum net upward force generated by our device is given by:

$$F = -\gamma L \cos\theta + \rho_w g A h_w \quad (1)$$

where $\gamma$ is the water surface tension coefficient, $L$ is the net contact length, $\theta$ is the contact angle between Parylene C and water, $\rho_W$ is the water density, $g$ is the gravity, $A$ is the EWP's flat area, and $h_w$ is the maximum deformation of the water surface before breaking. The value of $h_w$ relates to the contact angle between the EWP and the water surface, and consequently the buoyancy term is dependent on the surface tension. The dependence of $h_w$ on contact angle will be specified in equation (4), and the values of the constants are given in Supplementary Table 1. For our EWP design, equation (1) estimates that surface tension contributes ~25% of the net upward force, and surface tension induced buoyancy force accounts for the rest. We note that the buoyancy contribution becomes even more important than surface tension in heavier (>1 g) water striding robots because contact area grows faster than contact length as robot size increases. Our robot weighs 1.65 gram, and it can carry 1.44 gram of additional payload on the water surface. This additional payload allows the robot to paddle its legs (up to 10

Hz) without breaking the water surface (Supplementary Movie 4), which is crucial for robot locomotion.

This device further enables controllable and repeatable transitions through the water surface. We define transition controllability as the robot's ability to dive into water at a desired location and time. This form of controllability is absent in a previous study[24] that demonstrates transition by coating a microrobot with a surfactant. Furthermore, we are only concerned with sinking in shallow (<15 cm) and undisturbed water. Under these conditions and without control of the robot's pose during sinking, the robot always lands on its feet because its center of mass is lower than its geometric center. The EWPs initiate sinking with the electrowetting process—the modification of a surface's wetting properties under an applied electric field. When a voltage is applied to a conductive surface coated with a dielectric layer, there is a reduction of the contact angle between an electrolyte and the solid surface. This reduction of contact angle leads to two effects that enable sinking: surface tension reduction and spontaneous wetting.

First, the electrowetting process reduces surface tension by reducing the contact angle between the EWP's vertical walls and the meniscus surface (Fig. 3b). When a 600 V signal is sent to the EWP, an electric field perpendicular to the meniscus surface (parallel to the free water surface) is generated and it leads to a change of the contact angle governed by:

$$\cos\theta = \cos\theta_N + \frac{\epsilon_0\epsilon_1}{2\gamma d_H} V^2. \quad (2)$$

Here $\epsilon_0$ is the permittivity of free space, $\epsilon_1$ is the relative permittivity, $d_H$ is the dielectric coating thickness, $\theta_N$ is the nominal contact angle, and $V$ is the applied voltage. This reduction of contact angle reduces the upward surface tension force governed by the first term of equation (1). According to equation (2), a hydrophobic coating ($\theta_N > 90°$) can become hydrophilic ($\theta < 90°$) under a large voltage input, which changes the weight-bearing surface tension force to a downward pulling force.

The EWP's vertical walls are important for reducing surface tension because they lower the required operating voltage and mitigate the problem of dielectric breakdown. Without the

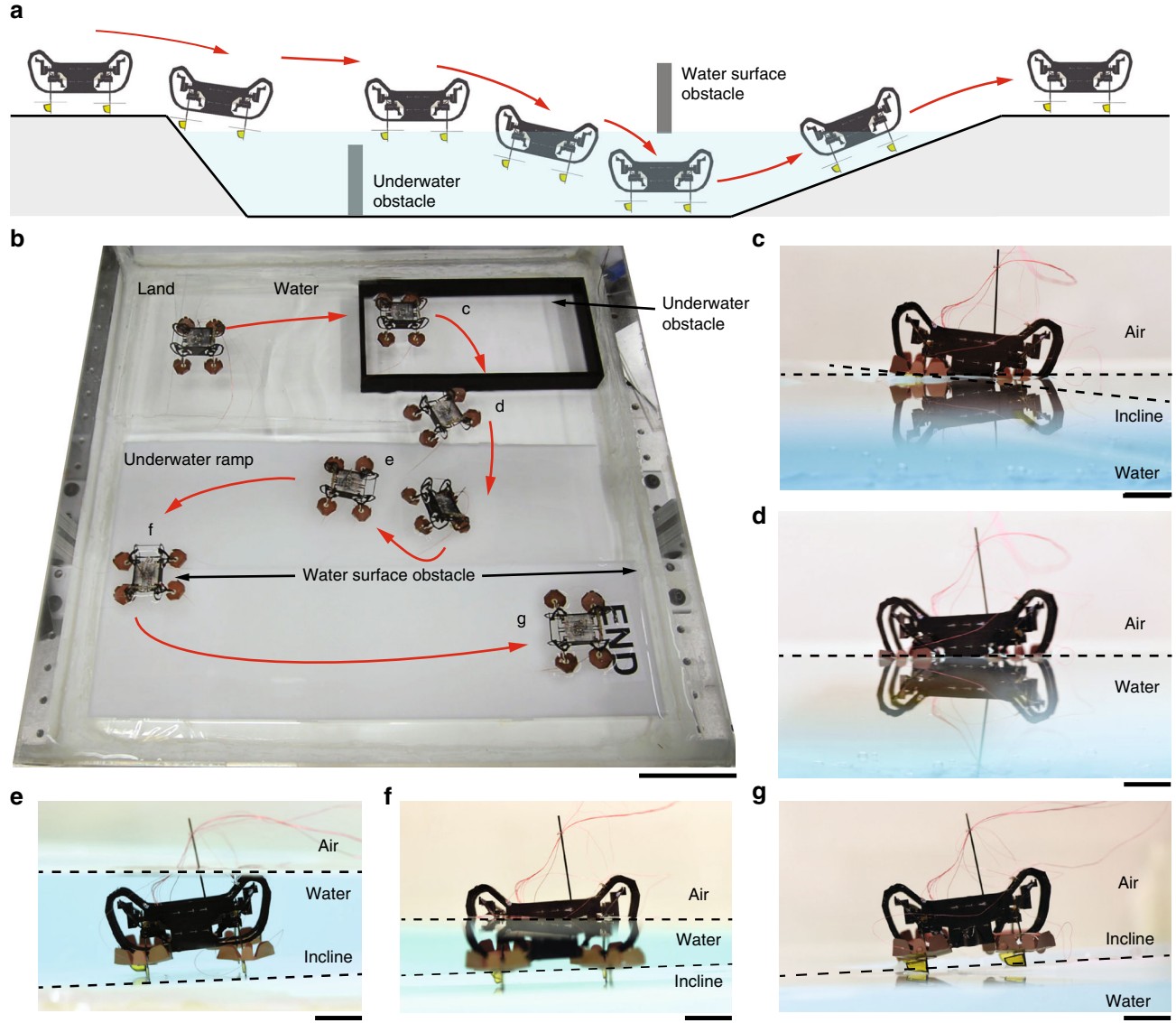

**Fig. 2** Robot demonstration. **a** An illustration of robot locomotion. The robot can walk on level ground, swim on the water surface, dive into water, walk underwater, and make transitions between ground, the water surface, and the underwater environment. **b** Top view composite image of the robot demonstrating hybrid locomotion described in **a**. Scale bar, 5 cm. **c** Side view of the robot walking down an incline and transitioning from land to the water surface. **d** The robot swims on the water surface. **e** The robot climbs an incline when it is fully submerged in water. **f** The robot gradually emerges from the air–water interface. **g** The robot completely exits water. Scale bars (**c**–**g**), 1 cm. In **c**–**g**, two drops of blue food coloring are added to deionized water to enhance the color of water in side view images

vertical walls, the fringing electric fields generated by the EWP's horizontal surface are much weaker. This requires a higher input voltage to achieve a similar reduction in contact angle, and can potentially cause dielectric breakdown of the insulating Parylene coating. Compared to a flat foot pad, the EWP's vertical walls strengthen the electric field between the device and the water surface meniscus (Fig. 3b) under the same input voltage. The conflicting relationship between contact angle reduction and dielectric breakdown is illustrated in Supplementary Figure 1a. The quadratic curve shows the required voltage that achieves a 100° contact angle reduction as a function of coating thickness, and the straight line shows the maximum EWP operating voltage before dielectric breakdown. The intersection of these lines predicts the minimum required coating thickness. To account for coating inhomogeneity of the fabrication process and the contact angle saturation effect,

we choose a coating thickness of 15 μm and an operating voltage of 600 V.

The height of the EWP's vertical sidewalls can be determined by analyzing the water meniscus profile[31]. The meniscus height (Fig. 3b) near the vertical sidewalls relates to the local contact angle:

$$h_v = \sqrt{2(1 - \sin\theta)}k^{-1}, \qquad (3)$$

where $k^{-1} = \sqrt{\gamma/\rho g}$ and it is defined as the characteristic length. This formula predicts the meniscus height to be ~3 mm. To ensure the local electric field is approximately perpendicular to the water meniscus, we set the EWP's sidewalls to be 4 mm tall, which is slightly larger than the meniscus height.

Following the reduction of surface tension, the buoyancy force also decreases due to change of $h_w$, which leads to spontaneous

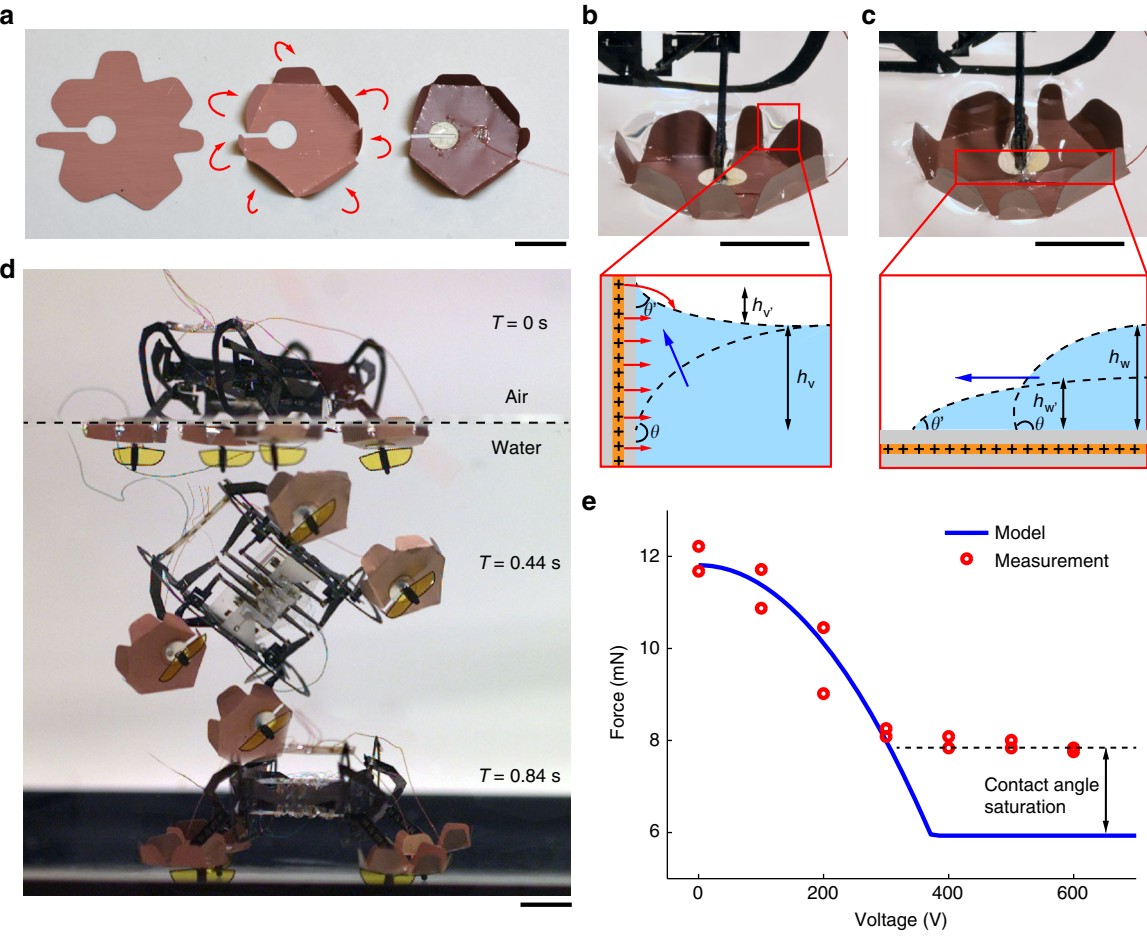

**Fig. 3** Electrowetting pad and controllable transition through the air-water interface. **a** Fabrication of an EWP. An EWP is laser machined from a 5 μm copper sheet, folded manually, wired, and then coated with 15 μm Parylene. **b** Modification of contact angle through electrowetting. When a 600 V signal is sent to the EWP, the contact angle between the EWP's vertical sides and the water surface decreases, which reduces the surface tension force. **c** Spontaneous wetting of the EWP's charged horizontal surface. The increase of surface wettability causes water to flow onto the EWP's upper surface, consequently sinking the robot. **d** Composite image of a robot sinking into water when all four EWPs are actuated with a 600 V signal. Scale bars (**a**–**d**), 5 mm. **e** Experimental characterization of the maximum upward force generated by an EWP at different voltages. Due to change of contact angle and spontaneous wetting, the net upward force decreases as the input voltage increases

wetting on the EWP's horizontal surface. Recall that $h_w$ is the maximum height difference between a static meniscus and a flat surface before the liquid spontaneously spreads on the surface (Fig. 3c). The relationship between $h_w$ and the surface contact angle is given by:

$$h_w = 2k^{-1} \sin \frac{\theta}{2}. \tag{4}$$

For the EWP design, $h_w$ reduces from 5 mm to 2 mm when the input voltage increases from 0 V to 600 V, which implies that the surface tension induced buoyancy force is reduced (second term of equation 1). Consequently, both surface tension and buoyancy force reduce and cause the robot to sink into water (Fig. 3d). Supplementary Figure 1b and Supplementary Movie 4 illustrate the spontaneous wetting process on a flat copper sheet coated with 15 μm of Parylene. In summary, charging the EWP's sidewalls reduces the surface tension force, and charging the EWP's horizontal surface lowers the buoyancy force.

We characterize the EWP performance by measuring the maximum surface tension force at different input voltages

(see Methods for experimental setup). Figure 3e compares the experimental measurement with the predicted values from equations (1), (2), and (4). The model shows good agreement with experiments for input voltages smaller than 400 V. The model underestimates the net upward force for input voltages higher than 400 V because it does not consider the contact angle saturation phenomenon[25], which is an experimental observation that no material can become completely hydrophilic regardless of the input voltage amplitude. In future studies, this discrepancy between model and measurement may be reduced by using alternative dielectric coatings that have smaller saturation angles.

Our experiments show that the maximum upward force an EWP generates is 11.5 mN. This force reduces to 8.2 mN when a constant signal of 600 V is sent to the device. This measurement is an upper bound of the EWP's performance as it is rigidly mounted on a force sensor. When installed on the mircorobot, the EWP's horizontal surface may not be parallel to the water surface due to fabrication imperfection, causing a reduction in contact area and maximum surface forces. As a result, we measure the maximum robot weight to be 3.09 g (65% of the maximum static measurement) before sinking.

**Locomotion on the water surface.** In addition to floating on the water surface, the robot is capable of locomotion including swimming forward and turning. Existing designs[14–20] based on water striders cannot be applied to our robot because stationary, long supporting legs inhibit walking on the ground. Instead, we require the robot to use the same set of actuators and legs to move on the water surface. This requirement imposes two major challenges: symmetric walking gaits for terrestrial locomotion cannot generate net propulsive force due to the time reversibility property of low Reynolds number flow, and the amplitude of the robot legs' swing motion, and thus the induced drag force, is substantially less than that of biological examples such as the

diving beetles (*Dytiscus marginalis*) and the diving flies (*Ephydra hians*).

Diving beetles swim underwater by paddling their hind legs asymmetrically (Fig. 4a) to generate unidirectional thrust[7]. During the power stroke, the hind leg tarsus and tibia flatten to maximize projected area and increase forward thrust. During the recovery stroke, the hind leg tarsus and tibia retract to minimize the projected area and reduce backward drag. Previous biomimetic studies analyzed the diving beetle's paddling leg trajectories and showed that they can be modeled by two serial links connected to each other and the body by two actuated rotational joints[32,33].

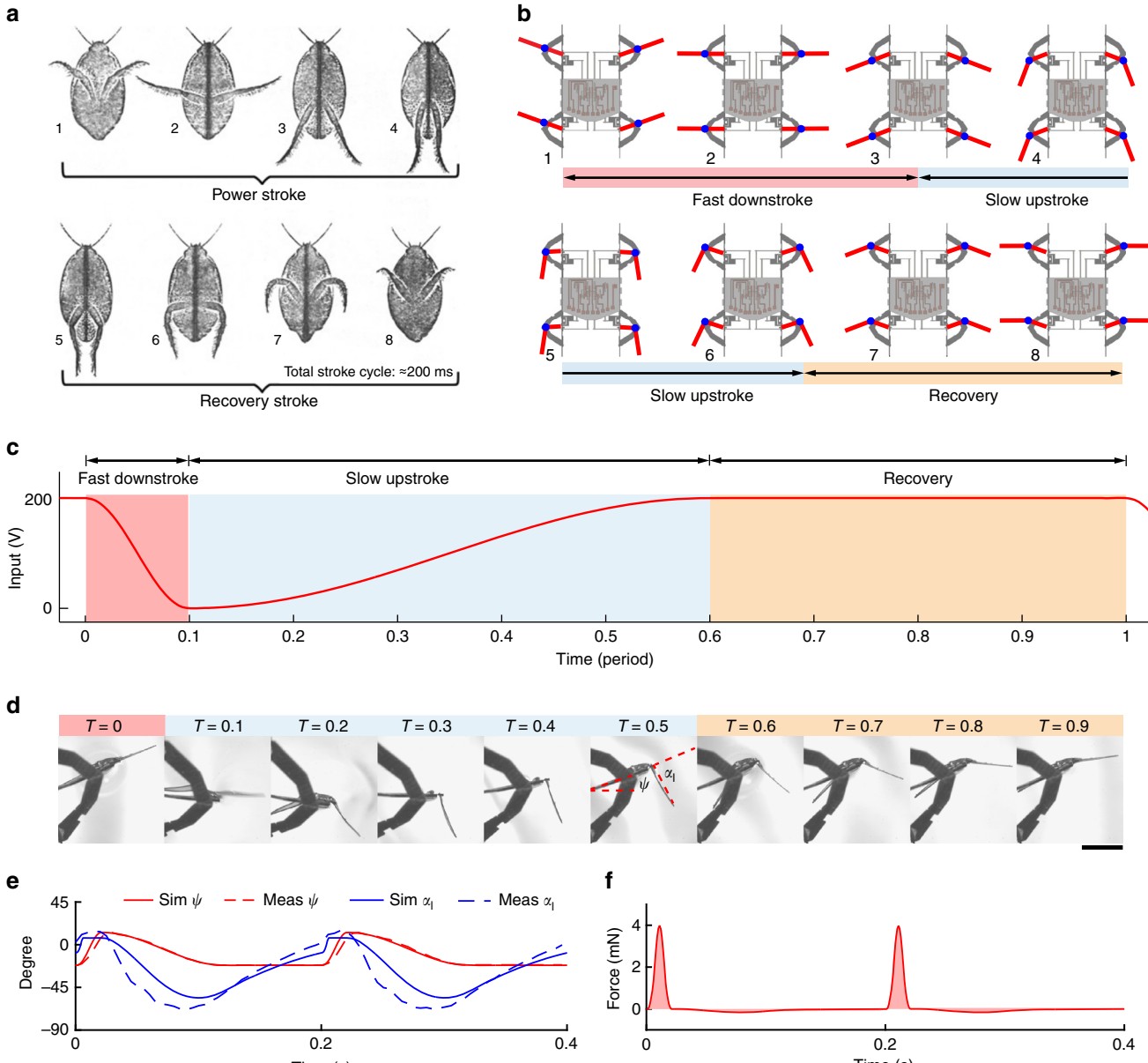

**Fig. 4** Aquatic flapping kinematics and dynamics. **a** Swimming behavior of a diving beetle. The power stroke and the recovery stroke are asymmetric (figure taken from[7]). **b** Bioinspired robot swimming kinematics feature asymmetric upstroke and downstroke without active control of the flap rotation. **c** Periodic control signal of the robot swing actuator is asymmetric. **d** Images of a single leg's swinging motion and the passive flap rotation in water. The images are taken 0.1 period apart, corresponding to the time scale of **c**. Asymmetric leg swinging motion leads to favorable passive flap rotation that increases net thrust. Scale bar, 5 mm. **e** Comparison of experimentally measured and simulated flapping motion $\psi$ and passive flap rotation $\alpha$. **f** Simulated instantaneous thrust force as a function of time. The experiments and simulations shown in **c**–**f** use the same control signal. **e**, **f** show that the quasi-steady model qualitatively agrees with the experimental result, and it predicts that an asymmetric driving signal generates larger net thrust force

Taking inspiration from the diving beetle's physiology and swimming mechanics, we develop passive swimming flaps[34] that generate asymmetric gaits for water surface locomotion. The robot leg and its flap constitute a two-serial links system: the leg motion is controlled and the flap rotation is passively mediated through an elastic joint. As shown in Fig. 1c, the flaps are fully submerged in water while the robot rests on the water surface. The flaps are designed to be passive devices that retract in a single direction. During the fast downstroke (Fig. 4b), the flaps remain fully open to generate forward thrust. During the slow upstroke (Fig. 4b), the flaps collapse to reduce drag. The flap rotation is passively mediated by forces from an elastic flexure, drag from the surrounding fluid, and the flap inertia. Consequently, developing an appropriate driving motion for the robot leg is crucial for achieving desired flapping kinematics[35].

To design the swimming kinematics and determine the flap area, inertia, and flexure stiffness, we conduct at-scale flapping experiments and construct quasi-steady, dynamical simulations (see Supplementary Notes 1–3 and Supplementary Figure 2). The kinematic parameters of stroke angle ($\psi$) and pitch angle ($\alpha$) are defined in the Supplementary Note 1, and they are labeled in Fig. 4d and Supplementary Figure 2c. During swimming, the robot's lift actuators are switched off and a piecewise sinusoidal driving signal is sent to the swing actuator (Fig. 4c). As shown in Fig. 4c, the fast downstroke occupies 10% of the flapping period, the slow upstroke takes 50% of the flapping period, and the actuator remains stationary for the remaining 40% of the time to allow the flap to return its nominal orientation.

Figure 4d shows a single leg flapping experiment using this driving signal at 5 Hz. The flap remains flattened during the fast downstroke ($T = 0$ to $T = 0.1$). At stroke reversal ($T = 0.1$), the flap begins to collapse while the leg slows down and reverses direction. This collapsing behavior is mainly due to the flap inertia and the force from the surrounding fluid. During the slow upstroke ($T = 0.1$ to $T = 0.6$), the flap remains collapsed to reduce drag. During the recovery phase ($T = 0.6$ to $T = 1$), the actuator is held stationary and the flap slowly rotates back to its nominal position due to the restoring torque from the flexure.

Figure 4e shows the tracked stroke angle ($\psi$) and the passive flap angle ($\alpha$), and superimposes the simulated stroke and flap motion based on the input signal from Fig. 4c. We observe qualitative agreement between the quasi-steady simulation and the experimental measurement. The error of the maximum predicted flap angle and phase offset are 4° and 6% period, respectively. As detailed in the supplemental material, this error arises primarily from ignoring the added mass effects and the collision between the flap and the central strut during the downstroke. We further estimate the drag force profile using the quasi-steady model. As shown in Fig. 4f, the thrust force is mainly generated during the downstroke, whereas the drag force during the upstroke and stroke recovery is small. Here the model estimates the time averaged force to be 0.13 mN from a single leg actuated at 5 Hz.

Driving all four robot legs with the same signal from Fig. 4c, we demonstrate robot forward swimming on the water surface. For the experiment shown in Fig. 5a, the robot swims 32 cm in 12 seconds, with an average speed of 2.7 cm s$^{-1}$ (0.7 body length (BL) per second). Figure 5b shows the instantaneous robot swimming speed extracted from a high-speed video of another swimming experiment (Supplementary Movie 3). The maximum and mean swimming speed are 8.1 (2.1 BL s$^{-1}$) and 2.8 cm s$^{-1}$ (0.7 BL s$^{-1}$), respectively. In addition to swimming forward, the

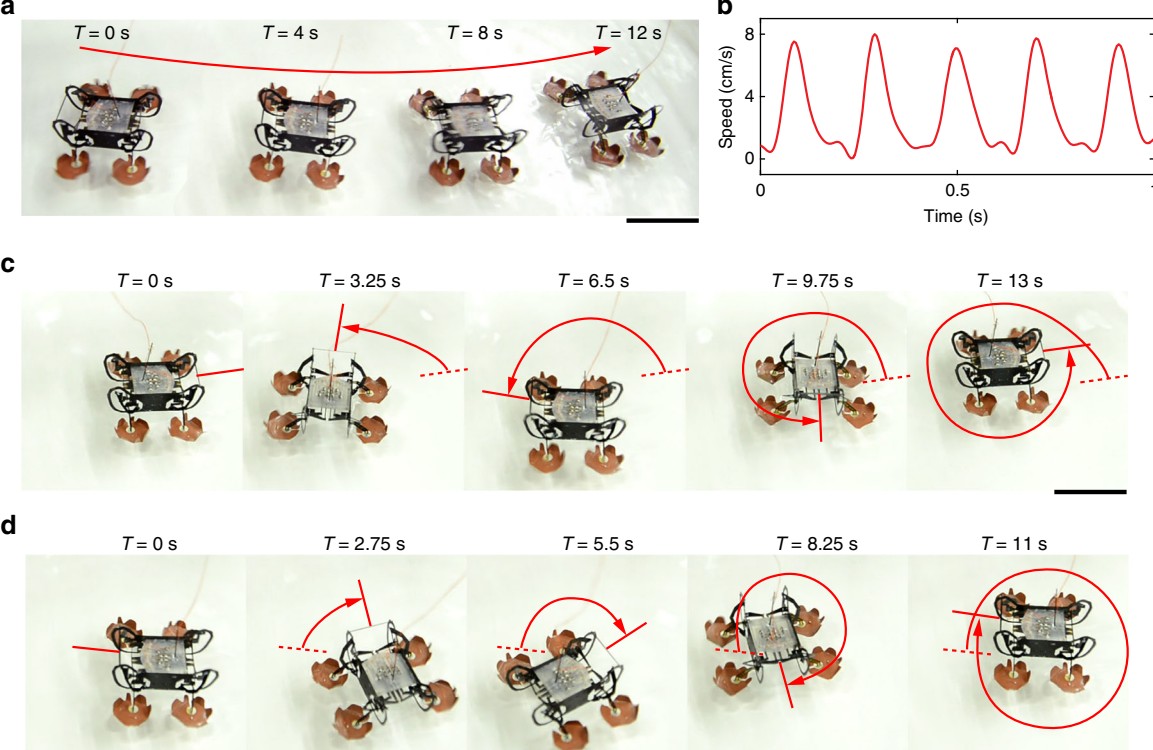

**Fig. 5** Robot swimming and turning on the water surface. **a** The robot moves on the water surface at 2.8 cm s$^{-1}$ with a 5 Hz swimming gait. **b** The robot's instantaneous swimming speed tracked using a high-speed video (Supplementary Movie 3). **c** The robot makes a complete left turn on the water surface in 13 seconds. **d** The robot makes a complete right turn on the water surface in 11 seconds. Scale bars (**a**, **c**, **d**), 2 cm. These demonstrations show that the robot can controllably move on the surface of water

robot can demonstrate left or right turns by turning off the actuators on the left or right side, respectively. Figure 5c, d, and Supplementary Movie 3 show the robot can make a complete right or left turn in 13 and 11 s, respectively.

We further compare robot locomotive efficiency in different environments by calculating the cost of transport:

$$c = \frac{P_{avg}}{mgv_{avg}}, \qquad (5)$$

where $P_{avg}$ and $v_{avg}$ are the net electrical power consumed by the robot and the average speed, respectively. The input power is calculated by measuring the voltage and current consumed by each actuator and then summing over all eight actuators for 20 periods:

$$P_{avg} = \frac{1}{T} \sum_{j=1}^{8} \int_{0}^{T} v_j(t) i_j(t) dt. \qquad (6)$$

Supplementary Table 2 lists the robot cost of transport for locomotion on land, underwater, on the water surface, and on an incline. The cost of transport for moving on the water surface is 18% higher than that of walking on land.

**Underwater to land transition**. Our microrobot is capable of walking and avoiding obstacles underwater (Supplementary Movie 5) and climbing up an incline to transition back onto land (Supplementary Movie 6). When it is fully submerged, the robot uses terrestrial walking gaits to demonstrate turning and walking. We estimate that buoyancy force accounts for 40% of the robot weight, which enhances locomotion and improves payload capacity underwater. In the aquatic-to-terrestrial transition process, there are two major challenges: overcoming the surface tension and climbing an incline.

As the robot moves through the air-water interface, the water surface exerts an inhibiting, downward force whose magnitude is approximately equal to the robot's weight. Although the robot can climb up an 11° incline underwater on surfaces covered by polydimethylsiloxane (PDMS), it is unable to break the water surface of while climbing up a 6° PDMS surface (Fig. 6a, and Supplementary Movie 7). Figure 6b shows the tracked robot front and hind leg trajectories: when completely submerged, the robot's front legs lift higher than its hind legs (red colored regions in Fig. 6b) due to the rearward location of its center of mass. Surface tension forces push down on the front of the robot as it moves out of water, inducing a clockwise torque with respect to the robot center of mass and causing the hind legs to lift higher than the front legs (blue colored regions in Fig. 6b). As the robot continues to climb upward, the body torque induced by surface tension becomes counterclockwise, causing the front legs to lift higher again (red colored regions in Fig. 6b). In this process, the downward surface tension force gradually increases and ultimately causes the robot hind legs to stick to the incline surface, preventing forward locomotion.

To quantify the magnitude of water surface tension force during the transition process, we measure the net force on a robot as it is slowly pulled out of water (Supplementary Fig. 3). As shown in Fig. 6c, the surface tension forces on the robot's circuit board and EWPs are approximately 16 mN and 9.3 mN, respectively. Although these forces are comparable to the robot's weight, previous work has demonstrated that HAMR can carry 2.9 g of additional payload[36] on flat surfaces.

These force measurements are conducted while mounting the robot parallel to the water surface, and consequently they do not account for the influence of the torque induced by the surface

tension force during the transition process. To quantify the influence of the surface tension torque, we pull the robot through the water surface on a 3° incline and quantify the deformation of the robot legs' transmission. As shown in the top view of Fig. 6d, the robot hind legs are splayed out further than the robot front legs, indicating that there is a larger force pushing down on the back of the robot. In an unloaded configuration (Fig. 1a), the robot's legs are approximately perpendicular with respect to the ground. In the configuration pictured in Fig. 6d, the robot hind legs splay outward 19° compared to the nominal leg orientation. This causes the back of the robot body to sag down and its front to tilt up. The side view in Fig. 6d shows that the robot body pitching increases to 14° ($\theta_B$) on a 3° ($\theta_I$) incline. This unfavorable body pitching $\theta_B$ exacerbates the adverse effects of climbing an incline, causing the robot's front legs to lift higher and preventing the robot's rear legs from lifting off the ramp surface.

We make two major modifications to mitigate the adverse effects caused by surface tension during underwater-to-ground transitions. First, the legs and swimming flaps are redesigned and fabricated monolithically using a 150 μm carbon fiber laminate, substantially reducing the deformation of the entire structure under load. In addition, we reduce the compliance of the legs' lift DOF by manually biasing the leg downward during the assembly process as detailed in a previous study[37]. This preloads the flexures to create a force bias that opposes gravity, effectively altering the transmission ratio to increase stiffness and reduce sagging. Second, we attach PDMS-coated foot pads to the robot's front legs to increase friction. The experiment illustrated in Fig. 6a, b shows that the robot's front and hind legs serve different functions during the transition process. The surface tension induced torque inhibits hind leg liftoff and increases friction on the hind legs. In contrast, the front legs experience lower normal and frictional forces which results in slipping. Attaching PDMS foot pads to the robot front legs increases friction on the front legs and reduces slipping.

Figure 6e shows a composite image of the robot climbing out of water on an acrylic, 3° incline at an average speed of 0.75 cm s$^{-1}$. Compared to the side view in Fig. 6d, this image shows that the modified robot does not pitch up noticeably during the transition process. Figure 6f further illustrates the corresponding front and hind leg trajectories. The lift motions of both front and back legs reduce as the robot feet emerge from the water surface. Once the robot front legs completely exit the water surface, the front legs' lift amplitudes recover. Due to surface tension force and torque, the cost of transport for transitioning from underwater to land is approximately four times larger than walking on level ground (Supplementary Table 2).

## Discussion

Our presentation of a hybrid terrestrial-aquatic microrobot includes a novel mesoscale device design that uses electrowetting to control surface tension magnitude and achieve controllable and repeatable water surface to underwater transitions, a multimodal strategy for locomotion on terrestrial domains and the water surface, and a detailed analysis of the challenges imposed by surface tension during a microrobot's transition from underwater to land. Our design satisfies the various constraints imposed by microrobot actuation, payload, and the different environments. Although many of these constraints lead to conflicting design requirements, they can be reconciled by leveraging physics, such as electrowetting or surface tension, unique to the millimeter scale. For instance, we design foot pads that rely on surface tension and surface tension induced buoyancy to reduce foot size. Furthermore, by leveraging electrowetting principles, we design four 25 mg EWPs that can both statically

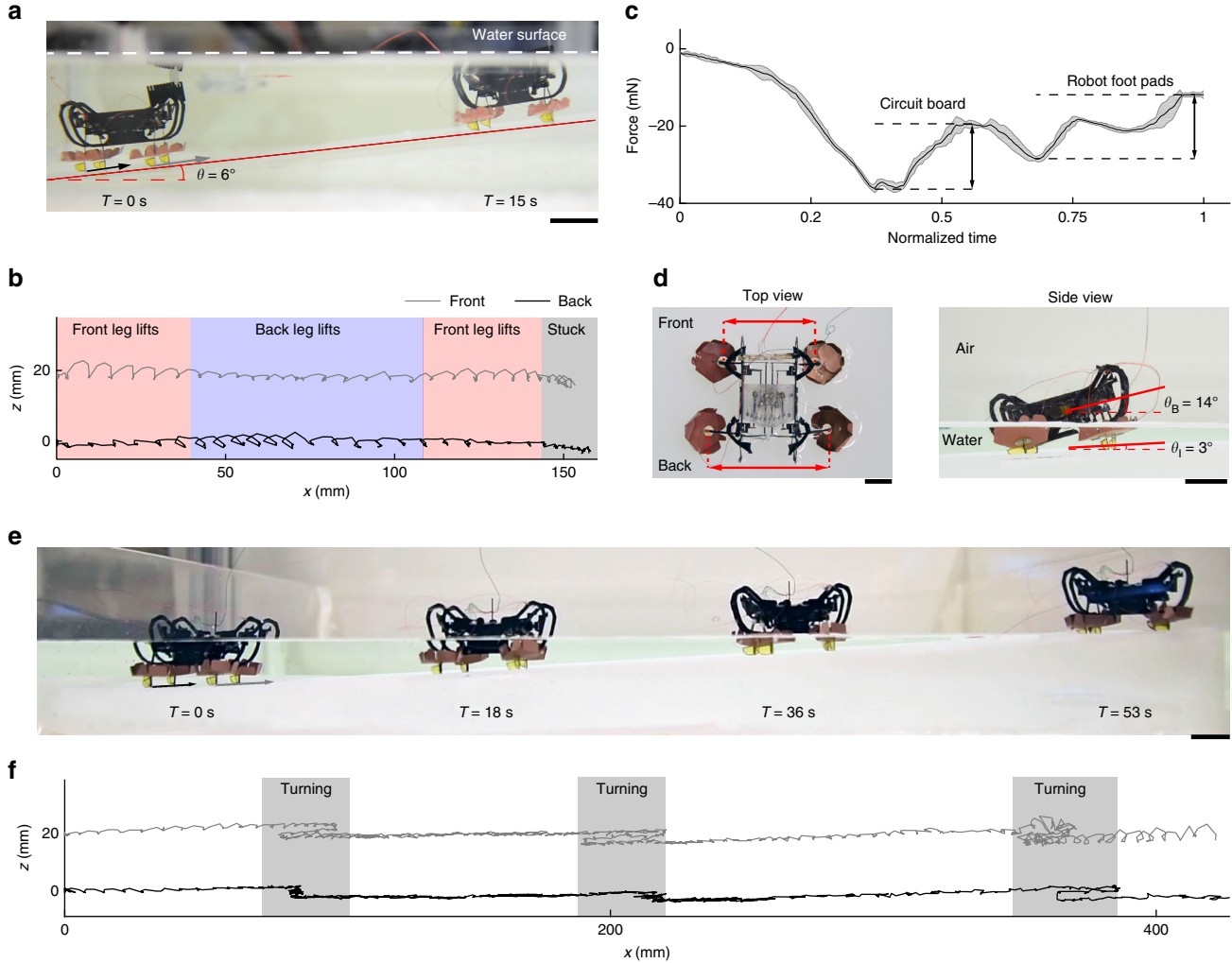

**Fig. 6** Robot water to land transition. **a** The robot is stuck at the air-water interface as it climbs an incline from underwater. The surface tension force exerts a counter-clockwise torque on the robot body, preventing the robot hind leg from lifting off. **b** The trajectories of the robot's front right and hind right legs. The ramp incline is subtracted from the trajectories to show leg lift at each step. When the robot is stuck, its hind legs cannot lift off the incline surface. **c** The net force on a robot as it is pulled out of water vertically. The net surface tension force exceeds the robot weight. **d** Top and side view images of a robot getting stuck at the air-water interface. The robot hind legs splay outward due to larger surface tension force on the rear of the robot. **e** After stiffening robot transmission in the lift DOF, the robot moves through the air–water interface on a 3° incline. **f** Robot leg trajectories during the water to land transition. The ramp incline is subtracted from the trajectories. During the transition process, the lift motion of both robot front and hind legs are reduced due to the inhibiting surface tension force. The robot leg motion recovers after the robot exits the surface of water. Scale bars (**a**, **d**, **e**), 1 cm

support 1.9 times the robot weight and sink the robot when actuated. Finally, whereas legged terrestrial locomotion involves discontinuous contact dynamics and friction, aquatic flapping locomotion at the low Reynolds number regime is continuous and requires asymmetric strokes. Using a combination of passive flaps and asymmetric driving, we develop a swimming strategy that has a similar cost of transport compared to robot terrestrial locomotion.

Compared to hybrid terrestrial-aquatic insects, microrobots have shortcomings in actuation and power density but also possess advantages in using engineered materials and electrostatic devices. To dive into water, a diving fly[4] can exert a downward force larger than 18 times its weight to overcome surface tension. Piezoelectric actuators cannot deliver such high force; however, a microrobot can utilize electrowetting to modify wettability —something that is unobserved among insects. In the case of swimming, the shortcomings in actuation can be compensated by exploiting fluid-structure interactions in passive mechanisms.

Whereas a diving beetle[7] can paddle with asymmetric power stroke and recovery stroke by independently controlling the motion of its tibia and tarsus[8], a microrobot can generate similar asymmetric paddling motion by merely controlling its leg (analogous to tarsus) swing. The flap (analogous to tibia) rotation is passively controlled through the coupling between the fluid flow, the flap inertia, and the elastic hinge to achieve efficient locomotion on the surface of water.

Further, this is a demonstration of a microrobot capable of performing tasks that are difficult for larger robots. To the best of our knowledge, no existing robot can walk on land, swim on the water surface, and transition between these environments. By leveraging surface effects that dominate at the millimeter scale, this work shows a microrobot can outperform larger ones in specific applications. In search and rescue missions, our robot has the potential to move through cluttered environments that are not accessible to larger terrestrial or aquatic robots.

Future studies can explore several topics to further improve microrobot locomotive capabilities in complex environments. Due to the lack of control during the sinking process, the current robot may flip over in presence of disturbances such as surface waves and dynamic flow underwater. Further, the robot cannot return to land without a modest ramp. These limitations can be addressed by enabling underwater swimming and improving climbing capabilities in hybrid terrestrial-aquatic microrobots. To demonstrate swimming, future research could involve designing meso-scale devices for buoyancy control[24], developing leg structures and associated gaits to generate lift force in addition to thrust, and conducting dynamical analysis to investigate underwater stability. To climb steeper incline or to return to land without a ramp, future research can incorporate electrostatic adhesion[13], gecko-inspired adhesives[38], or impulsive jumping mechanisms[24].

## Methods

**EWP fabrication**. The robot foot pad is fabricated from a 5 μm copper sheet. First, the copper sheet is cut into the desired pattern (Fig. 3a) using diode-pulsed solid state (DPSS) laser. Next, vertical walls around the foot pad base are manually folded by 90° under a microscope (Fig. 3a). Then we solder the foot pad using 51-gauge quadruple insulated wire and coat the device using Parylene C. The coatingprocess takes ~12 h to deposit a uniform layer of 15 μm Parylene C. Finally, the foot pad is attached to a 70 μm thick, circular fiber glass (FR4) piece (Fig. 3a). The fiber glass connection piece prevents the foot pad from shorting to the robot chassis.

**Experimental setup for measuring surface tension on the EWP**. An EWP is mounted on a capacitive force sensor and slowly pushed into water at ~0.2 mm s$^{-1}$ (Supplementary Fig. 1c). The net instantaneous force is measured by the force sensor. The red arrows in Supplementary Figure 1d, e indicate the difference between the minimum force and the net force once the EWP is completely submerged. This value represents the maximum upward force an EWP can generate. As shown in Supplementary Figure 1d, e, the net upward force reduces by 30% when a 600 V signal is sent to the EWP.

**Experimental setup for robot locomotion demonstration**. We built a 45 cm × 45 cm × 8 cm aquarium to conduct robot locomotion experiments in terrestrial and aquatic environments (Supplementary Fig. 4a). The aquarium is filled with deionized water at a depth of 4 cm. A 5 cm tall, 6° ramp is placed in the aquarium for the robot to walk from land to the water surface. An ~3 cm tall underwater obstacle is placed in the robot's swimming path. A 4 cm tall, 3° ramp is placed in the aquarium for the robot to climb out of water. Two cameras are placed above and on the side of the aquarium to take top and side view videos. The water in the aquarium is connected to electrical ground. The robot end-to-end locomotion experiments are conducted four times to demonstrate robot robustness and repeatability, and these trajectories are overlaid in Supplementary Figure 4b.

**Data availability**. The data and code that support the findings of this study are available from the corresponding author Y.C. upon reasonable request.

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

## Acknowledgements

We thank Sebastien de Rivaz and Yishan Zhu for comments and discussion. This work is supported by the Wyss Institute for Biologically Inspired Engineering. In addition, the prototypes were enabled by equipment supported by the Army Research Office DURIP program (award no. W911NF-13-1-0311).

## Author contributions

Y.C., N.D., and B.G. designed and fabricated the robot; Y.C., N.D., B.G., and H.W. conceived the experimental work; Y.C., N.D., and H.W. and R.J.W. contributed to modeling and data analysis; Y.C. wrote the paper. All authors provided feedback.

## Additional information

**Competing interests:** The authors declare no competing interests.

