## [Peer Review File · Nature Communications]

Reviewer #2 (Remarks to the Author):

"Controllable water surface to underwater transition through electrowetting in a hybrid terrestrial-aquatic microrobot" by Chen, Doshi, Goldberg, Wang, and Wood, describes the analysis and realization of a centimetric-scale robot that is capable of walking on water, underwater, on land, and — most impressively — transitioning among all three. The ability to transition from locomotion at the surface to underwater walking is novel; this feat presents an enormous challenge because, in order to dive, the robot must overcome the forces supplied by surface tension and buoyancy, the very same forces that allow the robot to walk on water in the first place. To accomplish this, the authors develop a clever mechanism that locally modifies the surface tension via an applied voltage which allows the robot to dive. To enable the robot to exit the water, the authors provide a careful analysis that provides constraints on the structural mechanical properties of the robot which allow it to overcome surface tension as it climbs an incline.

I found the paper to be well-written, novel, and interesting. The careful incorporation and exploitation of surface tension is well-done and extremely creative. I believe the paper will be of interest to a broad audience and I am happy to recommend it for publication in Nature.

A few minor points the authors may wish to consider:

1. In the introduction, "search and rescue" in environments with "puddles" reads a bit like the authors have a solution and they are looking for a problem. In my mind, the robot and the study are sufficiently interesting in and of themselves and the paper does not need what feels to me like an artificial justification. (Alternatively, the authors could expand on "search and rescue" and "puddles" to give the reader a more concrete example of what this might entail).
2. I have not heard the term "surface tension induced buoyancy." This term in equation (1) looks like a regular buoyancy term. Does "surface tension induced" imply that h_m is somehow set by the contact angle? (I'm not sure that is true; if the contact angle was 90 degrees, the buoyancy term could still support the robot if the walls on the foot were high enough ...) It would be helpful to add a sentence or two to clarify this terminology.
3. Small typo: in line 175, k is defined as a characteristic length but I think it should be $1/\text{characteristic length}$?
4. The supplementary movies are great!

Reviewer #3 (Remarks to the Author):

This paper demonstrates a quadrupedal microrobot that is capable of walking on level ground, paddling on the surface of water, sinking to the bottom of the water tank, and crawling up a mild incline to get back to land. One of the key results was the use of electrowetting pads (EWP) on the robot legs to allow electrical modulation of surface wettability, and thus enabled the robot to sink into the water when a voltage of 600V was applied. The application of electrowetting in robot locomotion environment transition is quite novel, and I suspect will be of interest to a broad group of readers. The authors also discussed in detail other design choices and modifications to address challenges and tradeoffs for the robot to transit between different locomotion environments. One of the design ideas was to equip the robot legs with passive, unidirectional flaps, to generate asymmetric strokes and allow the robot to use its terrestrial-based legs to effectively move forward in low Reynolds number fluid. Another modification was to increase the friction on the front legs, to mitigate the additional pitching issue caused by surface tension induced torque as the robot crawled across the water surface along the ramp from water to land. Force responses

during these interactions with different environments were carefully calculated and measured, and the authors provided detailed analysis of the comparison between these calculation and measurements. Improvements of designs were discussed based on the analysis. These design choices and the resulting improvement suggested that morphological features had a significant impact on locomotors' performance and efficiency when interacting with different locomotion substrates, and tradeoffs should be carefully considered for multimodal locomotor like the one shown in this paper.

Overall, it's a well-written, technically sound, and very interesting paper, and I would recommend this paper to be published after the comments and concerns being addressed.

My main concern is the limitation of the proposed electrowetting approach. The electrowetting approach only initiates the sinking behavior of the robot. Once the sinking is activated and robot is below the water surface, the position or the pose of the robot is no longer controllable underwater until the robot reaches the bottom of the tank. Therefore, depending on the wetting timing of the EWP, the robot could roll and flip over during the sinking even in relatively still water. In addition, the robot would not be able to transit back to land without a carefully designed ramp (up to approximately 6 degrees inclination for relatively high frictional surfaces, and up to approximately 3 degrees inclination for relatively low frictional surfaces). Such strict requirements could limit the possibility to extend the proposed approach to other platforms and environments.

I would suggest that the authors add a few sentences to discuss possible future directions to address these limitations that come with the instantaneous wetting approach, to help the readers better evaluate the possibility to apply such approach in different environments or explore additional design modifications. Without such discussions, I would suggest that the authors focus on the novel sinking behavior, and tune down their claims on the controllability during surface to underwater transition as well as the water to land transition. For example, in the title "controllable water surface to underwater transition", the word "controllable" is quite a strong claim, I would suggest that the authors either removed this description or be more specific what they mean. In addition, in several places, the paper claims that the robot can make controllable transitions between terrestrial and aquatic environments. The authors should make it clear what conditions are required for these transitions to work.

Minor comments:

1. Line 130 " θ is the contact angle between Parylene C and water"

Please mark θ on Fig. 1c or Fig. 3

2. Line 132 " h_m is the maximum deformation of the water surface before breaking (Supplementary Table 1)

h_m is not listed in Supplementary Table 1. Please add.

Is this parameter calculated or experimentally measured? The paper mentioned that this deformation induced buoyancy accounted 75% of the water supporting force for this robot, and this force was not utilized in previous water surface supporting devices. However, it's not clear if it's the EWP design or the robot light weight that allows this robot to effectively use this force. If the authors have a theoretical formula for this parameter, please provide it in the paper. If the parameter is experimentally measured, it will be helpful to add a few sentences to provide insights on what design modifications allows the generation of this force, and what constraints limit further increase of this force.

3. Line 251 "Figure 4e shows the tracked stroke angle (ψ) and the passive flap angle (α)"
Please define these two angles and mark them on Fig. 4b.

4. Line 292 "red colored regions in Fig. 4B"
Should be Fig. 6b

5. Line 315 "This unfavorable body pitching θ_B exacerbates the adverse effects"
 θ_B is not defined in the paper. Please define and mark it in Fig. 6d side view.

Reviewer #2 (Remarks to the Author):

“Controllable water surface to underwater transition through electrowetting in a hybrid terrestrial-aquatic microrobot” by Chen, Doshi, Goldberg, Wang, and Wood, describes the analysis and realization of a centimetric-scale robot that is capable of walking on water, underwater, on land, and — most impressively — transitioning among all three. The ability to transition from locomotion at the surface to underwater walking is novel; this feat presents an enormous challenge because, in order to dive, the robot must overcome the forces supplied by surface tension and buoyancy, the very same forces that allow the robot to walk on water in the first place. To accomplish this, the authors develop a clever mechanism that locally modifies the surface tension via an applied voltage which allows the robot to dive. To enable the robot to exit the water, the authors provide a careful analysis that provides constraints on the structural mechanical properties of the robot which allow it to overcome surface tension as it climbs an incline.

I found the paper to be well-written, novel, and interesting. The careful incorporation and exploitation of surface tension is well-done and extremely creative. I believe the paper will be of interest to a broad audience and I am happy to recommend it for publication in Nature.

We appreciate the reviewer’s encouraging comments on the novelty of our paper. The reviewer offers suggestions for further improvement of the manuscript. Our detailed response to the reviewer’s suggestion is given below.

A few minor points the authors may wish to consider:

1. In the introduction, “search and rescue” in environments with “puddles” reads a bit like the authors have a solution and they are looking for a problem. In my mind, the robot and the study are sufficiently interesting in and of themselves and the paper does not need what feels to me like an artificial justification. (Alternatively, the authors could expand on “search and rescue” and “puddles” to give the reader a more concrete example of what this might entail).

We agree with the reviewer that “search and rescue mission in environments with water puddles” is a far-fetched and limited application. In the second paragraph of the introduction, we want to convey the message that hybrid locomotion enables microrobots to accomplish tasks that are difficult for large robots. In the revision (second paragraph of the introduction), we remove the last sentence on search and rescue in presence of water puddles. Instead, we state the main message: these hybrid locomotion capabilities could potentially allow microrobots to explore diverse environment that are inaccessible to large robots.

2. I have not heard the term “surface tension induced buoyancy.” This term in equation (1) looks like a regular buoyancy term. Does “surface tension induced” imply that h_m is somehow set by the contact angle? (I’m not sure that is true; if the contact angle was 90 degrees, the buoyancy

term could still support the robot if the walls on the foot were high enough ...) It would be helpful to add a sentence or two to clarify this terminology.

We thank the reviewer for this comment and agree that the term “surface tension induced buoyancy” should be clarified. The second term in equation (1) is related to surface tension because h_m depends on the contact angle. The original paper may not be clear enough about the purpose of the side walls on the foot. The side walls are intended to generate an electric field parallel to the free water surface and hence strengthen the electrowetting effect (discussed in lines 159-174). The gaps between the walls permit water to flow onto the EWP’s horizontal surface. The red arrows in the attached figure 1a indicate the gaps, and the highlighted region in figure 1b shows where water can flow in.

Figure 1 | Gap between sections of wall of an EWP. **a** Water can flow onto the EWP’s horizontal surface through the gaps marked by the red arrows. **b** Gap between an EWP’s walls is marked by the red ellipses.

There is a small confusion in the original paper. The term h_m (in equation (1)) and the term h_w (in equation (4)) are identical. The parameter dependence on the contact angle is quantified in equation (4), and from equation (4) we predict that h_w changes from 5 mm to 2 mm under a 600V input. The parameter dependence is further illustrated in Fig. 3c and in Supplementary Movie 4. In the revision, we make sure the notation is consistent by changing h_m to h_w .

In summary, we clarify the term “surface tension induced buoyancy” by explaining the dependence of h_w on contact angle, and change h_m to h_w for consistency.

3. Small typo: in line 175, k is defined as a characteristic length but I think it should be $1/\text{characteristic length}$?

We thank the reviewer for pointing out the typo. In the revision, the expression is revised to $k^{-1} = \sqrt{\gamma/\rho g}$.

4. The supplementary movies are great!

We thank the reviewer for this encouraging comment on video quality.

Reviewer #3 (Remarks to the Author):

This paper demonstrates a quadrupedal microrobot that is capable of walking on level ground, paddling on the surface of water, sinking to the bottom of the water tank, and crawling up a mild incline to get back to land. One of the key results was the use of electrowetting pads (EWP) on the robot legs to allow electrical modulation of surface wettability, and thus enabled the robot to sink into the water when a voltage of 600V was applied. The application of electrowetting in robot locomotion environment transition is quite novel, and I suspect will be of interest to a broad group of readers. The authors also discussed in detail other design choices and modifications to address challenges and tradeoffs for the robot to transit between different locomotion environments. One of the design ideas was to equip the robot legs with passive, unidirectional flaps, to generate asymmetric strokes and allow the robot to use its terrestrial-based legs to effectively move forward in low Reynolds number fluid. Another modification was to increase the friction on the front legs, to mitigate the additional pitching issue caused by surface tension induced torque as the robot crawled across the water surface along the ramp from water to land. Force responses during these interactions with different environments were carefully calculated and measured, and the authors provided detailed analysis of the comparison between these calculation and measurements. Improvements of designs were discussed based on the analysis. These design choices and the resulting improvement suggested that morphological features had a significant impact on locomotors' performance and efficiency when interacting with different locomotion substrates, and tradeoffs should be carefully considered for multimodal locomotor like the one shown in this paper.

Overall, it's a well-written, technically sound, and very interesting paper, and I would recommend this paper to be published after the comments and concerns being addressed.

We appreciate the reviewer's encouraging comments on the novelty of our paper. The reviewer raises concerns about the controllability of sinking and points out potential limitations of aquatic-to-land transitions. In addition, the reviewer gives a number of minor suggestions. Our detailed response is given below.

My main concern is the limitation of the proposed electrowetting approach. The electrowetting approach only initiates the sinking behavior of the robot. Once the sinking is activated and robot is below the water surface, the position or the pose of the robot is no longer controllable underwater until the robot reaches the bottom of the tank. Therefore, depending on the wetting timing of the EWP, the robot could roll and flip over during the sinking even in relatively still water. In addition, the robot would not be able to transit back to land without a carefully designed ramp (up to approximately 6 degrees inclination for relatively high frictional surfaces, and up to approximately 3 degrees inclination for relatively low frictional surfaces). Such strict requirements could limit the possibility to extend the proposed approach to other platforms and environments.

We thank the reviewer for this comment. The reviewer raises a concern about robot stability underwater. In this study, we do not control the robot during sinking. Based on experimental observation, the robot always lands upright (~15 sinking experiments) without the need for

active control over orientation. This is because the sinking height is less than 15 cm, and the robot center of mass is lower than its geometric center. We agree that without control, the robot may flip over due to disturbances such as surface waves or flow under the water surface.

To ensure the robot can land upright under such disturbances, the robot needs to improve its stability property when fully submerged. One potential method is to add buoyant outriggers above the center of mass to ensure the robot is passively upright stable. Another future direction may involve demonstrating robot swimming in water. This can be achieved by re-designing the passive flaps to allow pitch rotation similar to that of a previous work [1]. We would then need to further re-design the leg driving trajectories to generate lift forces. We think enabling the robot to swim is a very interesting topic to explore, but the implementation and analysis are beyond the scope of this paper.

The second concern relates to the limitation on aquatic-to-terrestrial transitions. The reviewer points out that our robot can transition back to land via climbing a modest incline. This is problematic in more complex environments, such as in natural environments or situations with barriers.

In a previous work [1], the authors developed a device that can control microrobot buoyancy underwater by converting water into oxygen and hydrogen and collecting the gas. The authors have shown that a microrobot can exit water through igniting the gas and impulsive jumping. Similar strategies can be implemented in this robot to adapt to environments in which a modest incline is not present. In addition, it will be interesting to explore robot climbing on steep inclines, through methods including electrostatics [2], micro-spines [3], and gecko-inspired adhesives [4]. Again, these are very interesting directions for future research, but are beyond the scope of the current paper.

In summary, we agree that these limitations should be discussed in the paper, along with future directions. These discussions are added to the last paragraph of the revised manuscript.

I would suggest that the authors add a few sentences to discuss possible possible future directions to address these limitations that come with the instantaneous wetting approach, to help the readers better evaluate the possibility to apply such approach in different environments or explore additional design modifications. Without such discussions, I would suggest that the authors focus on the novel sinking behavior, and tune down their claims on the controllability during surface to underwater transition as well as the water to land transition. For example, in the title “controllable water surface to underwater transition”, the word “controllable” is quite a strong claim, I would suggest that the authors either removed this description or be more specific what they mean. In addition, in several places, the paper claims that the robot can make controllable transitions between terrestrial and aquatic environments. The authors should make it clear what conditions are required for these transitions to work.

We thank the reviewer for this comment. We agree that it is important to discuss the current limitations and future directions. In our response to the reviewer’s previous comment, we discussed current limitations and future directions including swimming, impulsive jumping out

of the water surface, and climbing. These discussions are added to the last paragraph of the manuscript.

In addition, the reviewer suggests that we define the phrase “controllable transition”. We agree with the reviewer that this is very important. Here, we use the word “controllable” to denote that the robot can transition from the water surface to underwater environments at a specified location and time. We do not mean the robot pose is controlled during the transition process. This clarification is made for water surface to underwater transitions. With respect to aquatic-to-terrestrial transitions, we mean that the robot is able to overcome surface tension and complete the transition. We feel that the word “controllable” is not necessary, hence this word is removed from the discussion on aquatic-to-terrestrial transitions. This clarification is added to the third paragraph in the section “Floating and controllable sinking through electrowetting”.

Minor comments:

1. Line 130 “ θ is the contact angle between Parylene C and water”

Please mark θ on Fig. 1c or Fig. 3

We thank the reviewer for this comment. In the revision, the contact angle θ is labeled in Fig. 3b and Fig. 3c. We further label the new contact angle θ' in response to voltage input.

2. Line 132 “ h_m is the maximum deformation of the water surface before breaking (Supplementary Table 1) h_m is not listed in Supplementary Table 1. Please add.

Is this parameter calculated or experimentally measured? The paper mentioned that this deformation induced buoyancy accounted 75% of the water supporting force for this robot, and this force was not utilized in previous water surface supporting devices. However, it’s not clear if it’s the EWP design or the robot light weight that allows this robot to effectively use this force. If the authors have a theoretical formula for this parameter, please provide it in the paper. If the parameter is experimentally measured, it will be helpful to add a few sentences to provide insights on what design modifications allows the generation of this force, and what constraints limit further increase of this force.

We thank the reviewer for raising this issue. There is a small inconsistency in our paper. The term h_m in line 132 and h_w in equation (4) are identical quantities. In the revision, h_m is changed to h_w for consistency. This parameter is calculated using equation (4). Our paper mentions that this deformation induced buoyancy accounts 75% of the net upward force, and it is estimated using the equation (1).

Substituting parameters from Supplementary Table 1, the two terms of equation (1) give:

$$F_{surf} = -\gamma L \cos \theta_N = 2.5mN, \text{ and } F_{bouy} = \rho_w g A h_w = \rho g A \left(2k^{-1} \sin \frac{\theta_N}{2} \right) = 7.5mN.$$

When no voltage is applied to an EWP, the value of the characteristic length k is 2.7 mm (based on equation (3)) and the value of h_w is 5 mm. In short, all these parameters are calculated and the

equations are already provided in the paper. We think the paper becomes clearer after h_m is changed to h_w for consistency. This correction is made in equation (1) of the revision.

Equation (1) shows that the surface tension force and surface tension induced buoyancy are applicable to small robots because they scale with either the contact length or the contact area. A large (>20 cm) and heavy (>20 g) legged robot in practice cannot use these effects because its weight scales with length cubed. The scaling of these two terms becomes important in the design of microrobots. For a robot that is the size (~1 cm) and weight (~10 mg) of a water strider, the buoyancy force is much smaller than the surface tension force. For a robot that is similar to our robot (~4 cm, ~ 1.5g), the buoyancy force will become more important than surface tension. The relative importance of these two terms relate to the microrobot size and weight. This discussion is added to the second paragraph of the section “Floating and controllable sinking through electrowetting”.

3. Line 251 “Figure 4e shows the tracked stroke angle (ψ) and the passive flap angle (α)”

Please define these two angles and mark them on Fig. 4b.

We thank the reviewer for the comments. These parameters are defined in the Supplementary Notes (section 1.2), and they are marked in Supplementary Figure 2c. Due to space constraints, we feel labeling them on Fig. 4b maybe too cluttered for the audience. In the revision, we add a sentence emphasizing that these parameters are defined and marked in the supplement (Section 1.2 and Supplementary Figure 2c). In addition, the angles are labelled in the revised Figure 4c.

4. Line 292 “red colored regions in Fig. 4B”

Should be Fig. 6b

We appreciate the reviewer for pointing out this typo. It is corrected in the revision.

5. Line 315 “This unfavorable body pitching θ_B exacerbates the adverse effects”

θ_B is not defined in the paper. Please define and mark it in Fig. 6d side view.

We thank the reviewer for this comment. In our original manuscript, the incline angle and the robot body angle are both labelled Θ . In the revision, we distinguish them by adding appropriate subscripts: Θ_B represents body pitch angle and Θ_I represents the incline angle.

References

- [1] Y. Chen, H. Wang, E. F. Helbling, N. T. Jafferis, R. Zufferey, A. Ong, K. Ma, N. Gravish, P. Chirarattananon, M. Kovac, R. J. Wood, A biologically inspired, flapping-wing, hybrid aerial aquatic microrobot. *Sci. Robot.* 2, eaao5619 (2017).
- [2] Yamamoto, A., Nakashima, T. and Higuchi, T., 2007, November. Wall climbing mechanisms using electrostatic attraction generated by flexible electrodes. In *Micro-NanoMechatronics and Human Science, 2007. MHS'07. International Symposium on* (pp. 389-394). IEEE.

[3] Asbeck, A.T., Kim, S., McClung, A., Parness, A. and Cutkosky, M.R., 2006, May. Climbing walls with microspines. In *IEEE ICRA*.

[4] Santos, D., Heyneman, B., Kim, S., Esparza, N. and Cutkosky, M.R., 2008, May. Gecko-inspired climbing behaviors on vertical and overhanging surfaces. In *Robotics and Automation, 2008. ICRA 2008. IEEE International Conference on* (pp. 1125-1131). IEEE.

Reviewer #2 (Remarks to the Author):

I am satisfied with the authors responses and revisions and I congratulate the authors on a very nice paper.

Reviewer #3 (Remarks to the Author):

The authors did a great job addressing my comments. The clarity and consistency of the paper are much improved. I don't have any further questions.

One small typo in the revised manuscript, line 243:

"... labelled in Figure 4c"

Should be Figure 4d

Response to reviewer comments

Reviewer #2 (Remarks to the Author):

I am satisfied with the authors responses and revisions and I congratulate the authors on a very nice paper.

We thank the reviewer for the positive feedback and the suggestions throughout the entire review process.

Reviewer #3 (Remarks to the Author):

The authors did a great job addressing my comments. The clarity and consistency of the paper are much improved. I don't have any further questions.

We thank the reviewer for the encouraging feedback. The reviewer's suggestions on the paper's Discussion section help us to further clarify the core findings.

One small typo in the revised manuscript, line 243:

"... labelled in Figure 4c"

Should be Figure 4d

This figure reference is changed to "Figure 4d" in the final revised manuscript.